# Network Pharmacology Analysis and Experimental Validation of Kaempferol in the Treatment of Ischemic Stroke by Inhibiting Apoptosis and Regulating Neuroinflammation Involving Neutrophils

**DOI:** 10.3390/ijms232012694

**Published:** 2022-10-21

**Authors:** Shan-Shan Zhang, Man Liu, Dong-Ni Liu, Yu-Fu Shang, Guan-Hua Du, Yue-Hua Wang

**Affiliations:** 1State Key Laboratory of Bioactive Substance and Function of Natural Medicines, Institute of Materia Medica, Chinese Academy of Medical Sciences & Peking Union Medical College, Beijing 100006, China; 2Beijing Key Laboratory of Drug Target Identification and New Drug Screening, Institute of Materia Medica, Chinese Academy of Medical Sciences & Peking Union Medical College, Beijing 100006, China

**Keywords:** kaempferol, ischemic stroke, network pharmacology, apoptosis, neuroinflammation, neutrophils

## Abstract

Kaempferol, a natural plant flavonoid compound, has a neuroprotective effect on ischemic stroke, while the specific mechanism remains unclear. In the current study, we applied the comprehensive strategy that combines network pharmacology and experimental evaluation to explore the potential mechanism of kaempferol in the treatment of cerebral ischemia. First, network pharmacology analysis identified the biological process of kaempferol, suggesting that kaempferol may partly help in treating ischemic stroke by regulating apoptosis and inflammatory response. Then, we evaluated the efficacy of kaempferol in the acute stage of ischemic stroke and elucidated its effects and possible mechanisms on cell apoptosis and neuroinflammation involved by neutrophils. The results showed that kaempferol could significantly reduce the modified neurological severity score (mNSS), and reduce the volume of cerebral infarction and the degree of cerebral edema. In terms of anti-apoptosis, kaempferol could significantly reduce the number of TUNEL-positive cells, inhibit the expression of pro-apoptotic proteins and promote the expression of anti-apoptotic proteins. Kaempferol may play an anti-apoptotic role by up-regulating the expression level of the BDNF-TrkB-PI3K/AKT signaling pathway. In addition, we found that kaempferol inhibited neuron loss and the activation of glial cells, as well as the expression level of the inflammatory protein COX-2 and the classic pro-inflammatory signaling pathway TLR4/MyD88/NF-κB in the ischemic brain, reduced MPO activity and neutrophil counts in peripheral blood, and down-regulated neutrophil aggregation and infiltration in the ischemic brain. Western blot revealed that kaempferol down-regulated the activation of the JAK1/STAT3 signaling pathway in neutrophils and ischemic brains. Our study showed that kaempferol inhibited the activation and number of neutrophils in the rat peripheral blood and brain, which may be related to the down-regulation of the JAK1/STAT3 pathway.

## 1. Introduction

Stroke is the most common cause of permanent disability and the third leading cause of death among adults worldwide [1]. Thrombolysis and interventional therapy can treat ischemic stroke by recanalization of the embolic cerebral blood vessels [2]. However, because of the narrow therapeutic window and the occurrence of ischemia-reperfusion injury (IRI), they can only provide limited clinical benefits. In addition, cell death is the end point of ischemic injury, while the existing therapies offer little protective effect against injury [3]. Therefore, neuroprotection is a popular direction for the development of anti-ischemic stroke drugs. Although the pathogenesis of ischemic stroke is complex, apoptosis and neuroinflammation are considered to be critical to the primary and secondary progression of brain injury. Ischemia reperfusion (I/R) can activate a variety of cell-death programs, such as necrosis, apoptosis or autophagy-associated cell death [4]. A series of studies have shown that neuronal apoptosis plays a leading role in delayed neuronal death after I/R [5,6]. Research has demonstrated that limiting the occurrence of apoptosis is very important for the development of neuroprotective agents to limit neuronal damage after acute cerebral ischemia [7]. Meanwhile, supporting evidence further suggests that anti-inflammatory therapy and immune deficiency can lead to better outcomes in ischemic stroke [8]. Post-stroke neuroinflammation is not only caused by microglia and astrocytes, but also by blood-derived white blood cells. Among the various white blood cells, neutrophils have recently aroused great interest and have been well-studied. Infiltrating neutrophils produce pro-inflammatory cytokines, matrix metalloproteinases (MMPs), nitric oxide (NO), reactive oxygen species (ROS) and other cytotoxic molecules that accelerate brain damage [9]. The release of MMPs then results in the breakdown of the blood-brain barrier (BBB) which plays a major role in maintaining homeostasis in the brain. The disruption of the BBB further promotes the infiltration of neutrophils from the blood circulation into the brain, aggravating brain injury and leading to high morbidity and mortality of cerebral ischemia reperfusion [10].

Kaempferol (3,5,7,4′-tetrahydroxy flavone) is a flavanol isolated from Kaempferia galanga L. that belongs to zingiberene family [11], which has been proved to show a certain neuroprotective effect in Alzheimer’s disease, Parkinson’s disease, ischemic stroke, epilepsy, major depression, anxiety and other neurological diseases [12]. Kaempferol has been found to have neuroprotective effects by reducing oxidative stress [13], neuroinflammation [14] and mitochondrial dysfunction [15] after brain IRI. However, it has not been reported whether kaempferol can play an anti-IRI effect by inhibiting apoptosis and regulating neutrophil functions, which aroused our great interest.

Therefore, in the present study, we first conducted a network pharmacological analysis, revealing that kaempferol could help in treating ischemic stroke, partly by modulating apoptosis and inflammatory response, and this effect may be related to the PI3K/AKT signaling pathway and STAT3 transcription factor. Then, we evaluated the efficacy of kaempferol in the acute stage of ischemic stroke and elucidated its possible mechanisms on cell apoptosis and neuroinflammation involved by neutrophils in the middle cerebral artery occlusion and reperfusion (MCAO) rats.

## 2. Results

### 2.1. Network Pharmacology Predicted the Biological Process and Enrichment Analysis of Kaempferol against Ischemic Stroke

Currently, to explore the multiple targets and various pathways of kaempferol against ischemic stroke, network pharmacology is performed. First and foremost, we predicted the possible targets of kaempferol and obtained 100 potential related targets after eliminating the duplicates. Meanwhile, 1854 ischemic stroke-related targets were collected from the integration of Online Mendelian Inheritance in Man (OMIM), DisGeNET, GeneCards and Therapeutic Target Database (TTD). Finally, a total of 43 common targets of kaempferol and ischemic stroke were obtained (Figure 1A), and they were input into the STRING database. Finally, a clearer protein–protein interaction (PPI) network diagram was depicted in Cytoscape software (version 3.7.2), including 36 main common nodes and 63 edges total after hiding disconnected nodes in the network (Figure 1B). At the same time, to reveal the functions of kaempferol in ischemic stroke treatment, we analyzed the biological process (BP), cellular component (CC) and molecular function (MF) terms of the common targets with the Database for Annotation, Visualization and Integrated Discovery (DAVID). In total, 146 GO terms were significantly enriched (*p* < 0.05), including 95 BP terms, 18 CC terms and 33 MF terms (Appendix A). The top 10 significantly enriched terms in the BP, CC and MF categories are displayed visually (Figure 1C). The results showed that the main BP terms were negative regulation of apoptotic process, response to drug, inflammatory response, cellular protein metabolic process and protein autophosphorylation; the main CC terms were plasma membrane, extracellular space, extracellular region, extracellular exosome and membrane; and the main MF terms were protein binding, ATP binding, enzyme binding, identical protein binding and serine-type endopeptidase activity.

Furthermore, we conducted a KEGG pathway enrichment analysis on 36 main common targets. Twenty-four pathways (*p* < 0.05) were identified (Appendix A). The top 10 signaling pathways were selected for visual display (Figure 1D). The results showed that the targets were enriched mainly in the Estrogen signaling pathway, Rap1 signaling pathway, PI3K-AKT signaling pathway, Ras signaling pathway and VEGF signaling pathway. Most of the enriched signaling pathways were associated with regulation of cell function, indicating that kaempferol may treat ischemic stroke by regulating the functions of various central and peripheral cells. In addition, the transcription factor specific protein 1 (SP1), nuclear factor of kappa light polypeptide gene enhancer in B-cells 1 (NF-κB1) and signal transducer and activator of transcription 3 (STAT3) may be key transcription factors in the treatment of ischemic stroke with kaempferol (Figure 1E).

### 2.2. Effect of Kaempferol against Transient Focal Cerebral I/R Injury

When compared to the I/R group, kaempferol 100 mg/kg treatment reduced the infarct volume (Figure 2A,B) and brain water content (Figure 2C) in experimental cerebral ischemic stroke rats, while treatment with kaempferol 100 mg/kg improved body weight and neurological function recovery in the acute phase of cerebral ischemia (Figure 2D,E).

### 2.3. Kaempferol Attenuated Cell Apoptosis in Cerebral I/R Rats

TUNEL staining was used to observe the effect of kaempferol on I/R-induced apoptosis in the penumbra, ischemic core and striatum (location shown in Appendix A). There were no apoptotic cells in the sham group, while in the model group the number of TUNEL-positive cells was significantly increased. It is noteworthy that kaempferol administration almost completely inhibited the TUNEL fluorescence signal induced by I/R (Figure 3A,B).

Moreover, it is well known that the activation of caspase-3 is necessary for the execution of apoptotic events [16]. Consistent with TUNEL staining results, compared with the sham group the expression of caspase-3 protein in the model group increased significantly, and kaempferol administration significantly reversed this change (Figure 3C). Bax is a pro-apoptotic protein, while Bcl-2 and Bcl-xl are important anti-apoptotic proteins. The ratio between Bcl-2/Bax proteins is a key indicator of apoptosis initiation. The results showed that the ratio between Bcl-2/Bax and the expression level of Bcl-xl in the ischemic cortexes of the I/R group decreased significantly, while kaempferol administration obviously increasedthe levels of these indexes. The above results proved that kaempferol had a significant anti-apoptotic effect. To verify the results of KEGG enrichment analysis, we further detected the expression and phosphorylation levels of the apoptosis-related signaling pathway BDNF-TrkB-PI3K/AKT by Western blot. As we expected, the expression levels of the BDNF and TrkB and the phosphorylation levels of the PI3K and AKT proteins were prominently reduced in the model group compared with the sham group, while kaempferol administration significantly increased the expression levels or phosphorylation levels of these proteins (Figure 3F–I).

### 2.4. Kaempferol Reduced Neuron Loss as Well as the Activation of Microglia and Astrocytes in Cerebral I/R Rats

NeuN is a marker of neuronal cells. Immunofluorescence results showed that clear NeuN staining was observed in the sham group, while NeuN-positive cells almost disappeared in the penumbra, ischemic core and striatum (the location shown in Appendix A) of rats in the I/R group, indicating that a large number of neuronal cells were lost on the third day after I/R. However, compared with the I/R group, the administration of 100 mg/kg kaempferol significantly increased the number of NeuN positive cells in the ischemic brain (Figure 4A). Activation of glial cells is one of the hallmarks of neuroinflammation, including changes in the number, size and shape of glial cells. Iba-1 and GFAP, specific markers of microglia and astrocytes, respectively, were used for immunofluorescence staining in this study. The results showed that compared with the sham group, the size and number of microglia and astrocytes in the penumbra, ischemic core and striatum (the location shown in Appendix A) increased in I/R group. Administration of 100 mg/kg kaempferol significantly decreased the size and number of both glial cells, indicating that kaempferol could reduce neuroinflammation and I/R injury by inhibiting the activation of microglia and astrocytes (Figure 4B,C), and the effect of kaempferol on reducing neuronal cell loss may also be related to its inhibition of glial activation.

Finally, we examined the effects of kaempferol on the expression of inflammatory protein COX-2 and the classical inflammatory signaling pathway TLR4/MyD88/NF-κB. The results showed that compared with the sham group, the expression levels of the COX-2, TLR4, MyD88 protein and the phosphorylation level of the NF-κB protein in the ischemic cortex of rats in I/R group increased (Figure 4D–G). Compared with the I/R group, however, kaempferol administration at 100 mg/kg significantly decreased the expression level or phosphorylation level of these proteins, indicating that kaempferol has strong anti-neuroinflammatory activity.

### 2.5. Kaempferol Alleviated Blood Brain Barrier Disruption in Cerebral I/R Rats

BBB permeability was assessed by Evans blue leakage. Our results showed that Evans blue leakage was significantly increased in the I/R group compared with the sham group; administration of kaempferol reduced Evans blue leakage to the level of the sham group, and 100 mg/kg kaempferol treatment showed the best performance (Figure 5A,B). We further examined the effects of kaempferol on the expression level of tight junction proteins ZO-1 and occludin. Compared with the sham group, the expression levels of the ZO-1 and occludin proteins in the ischemic cortex of rats in the I/R group significantly decreased, while 100 mg/kg kaempferol significantly increased the expression levels of these two proteins (Figure 5C,D).

### 2.6. Kaempferol Inhibited Neutrophil Activation, Aggregation and Infiltration in Cerebral I/R Rats

The peripheral blood and serum of I/R rats were collected after 24 h reperfusion in order to evaluate the effects of kaempferol on the activation of neutrophils in peripheral blood. Compared with the sham group, the number of neutrophils and MPO activity in the peripheral blood of rats increased after I/R. Treatment with 100 mg/kg kaempferol reversed the increase in neutrophil count (Figure 6A) and MPO activity (Figure 6B) induced by I/R.

Activated neutrophils aggregate at the site of ischemic injury and release ROS, proteases, chemokines and cytokines, initiating a self-activating inflammatory cascade. In order to evaluate the effect of the immune regulation of kaempferol (100 mg/kg, i.g.) administered 2 h immediately after reperfusion, the mRNA of the ischemic cortex was isolated 3 days after MCAO, and the expression of inflammatory markers was analyzed by qPCR. Strikingly, the transcription levels of multiple pro-inflammatory factors including TNF-α, IL-1β and IL-6 were significantly down-regulated and the anti-inflammatory factor IL-10 was significantly up-regulated in the stroke lesions of kaempferol-treated rats (Figure 6C). The above results further confirmed that kaempferol has a certain anti-inflammatory effect. Of particular interest, the mRNA expression of neutrophil attracting chemokines significantly decreased in the kaempferol-treated group such as chemokine (C-C motif) ligand 5 (CCL5), chemokine (C-X-C motif) ligand 1 (CXCL1), CXCL9 and CXCL10. Collectively, these results illustrated that kaempferol significantly attenuated neutrophils’ aggregation in stroke lesions following cerebral I/R insult.

In addition, neutrophils can be easily identified by their multi-lobed nuclei after HE staining [17], thus reflecting the degree of neutrophil infiltration into the ischemic brain. In the sham group, neutrophils were rarely detected throughout the brain, while in the vehicle-treated I/R group, there was a large number of infiltrating neutrophils in the penumbra and ischemic core of the cortex as well as striatum (location shown in Appendix A). However, a 100 mg/kg kaempferol treatment significantly reduced the infiltration of neutrophils into the ischemic brain (Figure 6D).

Finally, the expression levels of neutrophil-related proteins including citrullinated histone H3 (CitH3), Galectin-3 and intercellular adhesion molecule-1 (ICAM-1) were analyzed by Western blotting. The results showed that compared with the sham group, I/R injury resulted in a significant increase in the expression of CitH3 (Figure 6E), Galectin-3 (Figure 6F) and ICAM-1 (Figure 6G), while a 100 mg/kg kaempferol treatment significantly reduced the expression levels of these proteins.

### 2.7. Kaempferol Down-Regulated the Expression of JAK1/STAT3 Signaling Pathway in Peripheral Blood Neutrophils and Ischemic Cortex

It has been established that the signal transducers and activators of the transcription (STAT) family play an important role in neutrophil function [18]. Therefore, we detected the effect of kaempferol on the expression of the JAK1/STAT3 signaling pathway in the peripheral blood neutrophils and ischemic cortex of I/R rats by Western blotting. We found that cerebral I/R significantly increased the levels of JAK1 and STAT3 on the protein expression as well as the level of STAT3 on the protein phosphorylation in peripheral blood neutrophils (Figure 7A–C). Western blot assay results also showed that the protein expression of JAK1 as well as protein phosphorylation of JAK1 and STAT3 in the ischemic cortex of rats were markedly increased after cerebral I/R (Figure 7D–F). However, the administration of 100 mg/kg kaempferol observably reduced the expression levels or phosphorylation levels of these proteins.

## 3. Discussion

In this study, we conducted a systemic study using a combination of network pharmacology and experimental verification to elucidate the pharmacological action and possible therapeutic mechanisms of kaempferol on cerebral ischemic stroke. Firstly, according to the results of the GO enrichment analysis, the biological process of kaempferol in the treatment of ischemic stroke mainly includes the negative regulation of apoptosis and inflammation. In addition, according to the results of the KEGG pathway enrichment analysis, kaempferol mainly interferes with the occurrence and development of ischemic stroke through the signal pathways such as Estrogen, Rap1 and PI3K-AKT, which are mainly related to the regulation of cell function. In conclusion, the anti-apoptotic and anti-inflammatory effect, especially the inflammatory process involving neutrophils, needs further exploration. Therefore, considering the prediction results of network pharmacology, and the fact that almost no research has been conducted to explore whether kaempferol can regulate the apoptotic process and neuroinflammation involved by neutrophils after ischemic stroke, we decided to perform biological experiments on kaempferol to fill the gap in this area.

Firstly, we used the well-established MCAO rats model to study the neuroprotective effect of kaempferol. We demonstrated that kaempferol has a strong protective effect on the acute phase of brain I/R injury, which is consistent with previous reports [14,19]. Specifically, in the rat model of brain ischemia induced by MCAO, the therapeutic administration of 100 mg/kg kaempferol significantly reduced the infarct volume and brain edema, and improved neurological deficit and weight recovery.

Subsequently, we confirmed the protective effect of kaempferol on cell apoptosis after cerebral ischemia-reperfusion, especially in the penumbra, the most salvageable area after ischemic stroke, where the cells mainly die of apoptosis after ischemia [20,21]. Therefore, our results showed that kaempferol could protect against cerebral IRI by inhibiting cell apoptosis.

In terms of mechanism, Cytochrome c (Cytc) released after the opening of mitochondrial permeability transition pores (MPTPs) leads to endogenous apoptosis [22], while exogenous apoptosis is mediated by cell-surface death receptors. Both eventually activate caspase proteins such as caspase-3, resulting in nuclear DNA damage and cell apoptosis [23]. The occurrence and development of cell apoptosis is also regulated by various genes, especially those closely related to the Bcl-2 family. Bcl-xl is another anti-apoptotic protein that inhibits the release of Cytc, endonuclease G, and apoptosis-inducing factor (AIF) from mitochondria [24], while during cerebral ischemia, the pro-apoptotic protein Bax is activated and transferred to the mitochondria, resulting in increased mitochondrial membrane permeability and subsequent Cytc release [25]. Our results showed that cerebral I/R increases the expression level of caspase-3 and Bcl-xl as well as the ratio of Bax to Bcl-2, while administration of 100 mg/kg kaempferol for 3 days decreased the expression levels of caspase-3 and Bax proteins and increased the expression levels of Bcl-2 and Bcl-xl proteins in ischemic penumbra, resulting in an increase of the Bcl-2 to Bax ratio and the inhibition of the occurrence of apoptotic events after cerebral ischemia-reperfusion.

BDNF-TrkB-PI3K/AKT is a classic anti-apoptotic signaling pathway, and activated AKT inhibits apoptosis by phosphorylating its substrate, such as the Bcl-2-associated death protein [26,27,28]. BDNF is widely distributed throughout the brain and has been reported to inhibit caspase-3 activity, increase the expression of the Bcl-2 protein and inhibit intracellular calcium overload, thereby reducing ischemic brain injury [5,28,29]. Our results suggest that kaempferol significantly increased the expression levels of the BDNF and TrkB proteins as well as the phosphorylation levels of the PI3K and AKT proteins after cerebral IRI, indicating that the neuroprotective effect of kaempferol may be involved in the activation of the BDNF-TrkB-PI3K/AKT signaling pathway.

Modulating post-stroke neuroinflammation has been suggested as an important therapeutic strategy in cerebral ischemia. Our previous investigation identified that kaempferol exerted protective effects on neuroinflammation and BBB destruction in the cortex [30] and striatum [31] of mice injured by LPS. In addition, kaempferol resisted cerebral IRI by inhibiting the activation of microglia, the overproduction of pro-inflammatory cytokines and the expression of inflammatory signaling pathways in the subacute phase [14]. Recently, neutrophils, blood-derived cell components, have received extensive attention. Under normal physiological conditions, neutrophils participate in innate immune responses against invading pathogens [8]. However, after pathological diseases such as ischemic stroke, neutrophils destroy the functions of endothelial cells and microglia, and promote tissue damage by releasing various inflammatory mediators [32]. However, the effect of kaempferol on neuroinflammation in the acute stage after ischemic stroke, especially neutrophil-involved neuroinflammation, has not been explored to a great extent.

Previous studies have shown that microglia are the major innate immune cells in the central nervous system (CNS), regulating brain development, homeostasis and neuroinflammation in disease and aging. Its activation will not only lead to the release of pro-inflammatory cytokines such as TNF-α, IL-1β and IL-6, but also lead to the overexpression of neurotoxic mediators such as iNOS, NO, COX-2 and ROS, thus further aggravating neuronal death and brain injury [33]. Similarly, activated astrocytes produce cytokines such as IL-1, IL-6, IFN-γ and TNF-α and interact with inflammatory cells [34]. Our results showed that administration of 100 mg/kg kaempferol significantly inhibited the activation of microglia and astrocytes in the penumbra, ischemic core and striatum, and alleviated the neuroinflammatory response induced by cerebral I/R. We also found that kaempferol could significantly inhibit neuronal cell loss in the ischemic brain, which may be related to the inhibition of glial activation. COX-2 mainly catalyzes the conversion of arachidonic acid to neurotoxic prostaglandin E2 (PGE2), promoting the release of inflammatory mediators and the occurrence of neurodegenerative diseases [35]. After binding to the ligand, TLR4 activates downstream NF-κB, translocates it to the nucleus, initiates the expression of inflammatory factors and ultimately causes inflammation [36]. Our results suggested that kaempferol may reduce neuroinflammation by reducing the expression of the inflammatory protein COX-2 and down-regulating the expression level of the TLR4/MyD88/NF-κB pathway.

The BBB plays a vital role in maintaining CNS homeostasis. The disruption of the BBB results in blood components such as neutrophils spilling into the brain, followed by impaired normal neuronal function. At the same time, neutrophils contribute to this disruption through the release of proteases (MMPs, elastase, cathepsin G and proteinase 3), ROS and during the process of migrating across the cerebral endothelium [37]. Tight junctions (TJs) are important structural and functional components of the BBB, which are composed of the cytoskeleton and TJ proteins. The breakdown of TJ proteins, such as ZO-1 and occludin, helps the BBB breakdown and the development of cerebral edema. The results showed that kaempferol significantly reduced EB leakage and increased the expression levels of ZO-1 and occludin. The results suggest that kaempferol may protect BBB damage after I/R and thus minimize subsequent infiltration of neutrophils into the ischemic brain.

Under physiological conditions, a high number of neutrophils are stored in the marginated pool, while IRI could induce the neutrophil release from the marginated pools to the circulating pools, resulting in an increase in the number of neutrophils in the peripheral blood [2]. MPO, as a neutrophil marker, is used to characterize the degree of neutrophil proliferation and activation [38]. Previous studies have shown that kaempferol can alleviate disease symptoms by inhibiting neutrophil infiltration or the activity of MPO in the high-fat-diet-induced obesity model [39], the lung I/R injury model [40], the keratitis model [41] and the gastric mucosal injury model [42]. Our results showed that kaempferol significantly reduced the number of neutrophils in peripheral circulation and MPO activity, thus reducing I/R-induced peripheral neutrophils’ activation.

Subsequently, neutrophils accumulated in ischemic lesions, releasing a large number of proteases, chemokines, cytokines and ROS, which subsequently initiate a self-activated inflammatory cascade. Proteases such as neutrophil elastase (NE), MMP-9 and MMP-3 can degrade the basement membrane and extracellular matrix, increase BBB permeability and brain edema, and aggravate ischemic injury [37]. Chemokines, together with adhesion molecules, are postulated to be important factors causing leukocyte infiltration into CNS [43]. In addition, elevated levels of pro-inflammatory cytokines including TNF-α IL-1β and IL-6 are associated with the breakdown of the BBB and recruitment of neutrophils into the ischemic brain [44], while the anti-inflammatory cytokine IL-10 inhibits the production of cytokines by neutrophils [45]. RT-PCR showed that kaempferol observably reduced protease including MPO, NE, MMP-9, MMP-3 and Galectin-3; chemokines including CXCL1, CXCL9, CXCL10, CCL5, and pro-inflammatory cytokines including TNF-α, IL-1β and IL-6 released by accumulated neutrophils. After treatment with kaempferol, the transcription levels of these factors decreased significantly and the number of infiltrating neutrophils into the ischemic brain decreased remarkably. At the same time, the expression levels of neutrophil-related proteins including CitH3, Galectin-3 and ICAM-1 were significantly reduced. Among them, CitH3 and Galectin-3 showed the most significant expression changes. As a marker of neutrophil extracellular traps (NETs) formation, CitH3 is combined with elastase, cathepsin G, MMP-9, MPO and depolymerized DNA to form NETs and promote thrombosis [46]. Galectin-3 is a neutrophil activator, which can promote the recruitment of neutrophil to the lung in a murine model of lung infection by Streptococcus pneumoniae [47]. However, for other reported neutrophil related proteins such as protein arginine deiminase 4 (PAD4), iNOS, integrin β2, MMP-2 and MMP-3, kaempferol could not restore the expression level to normal (western blot results shown in Appendix A). This further suggested that although the inhibitory effect of kaempferol on neutrophils in this study was significant, it was not comprehensive.

STAT3 is a transcription factor enriched by TRRUST analysis, and the JAK/STAT signaling pathway is related to inflammatory response, oxidative stress, cell injury and apoptosis [48] and can be activated by many pro-inflammatory cytokines such as IL-6 and IL-11 released from the ischemic brain after ischemic stroke [49]. STAT3 has a crucial role in emergency granulopoiesis and mature neutrophil function [50], and is highly involved in neutrophil differentiation, chemotaxis and migration [51,52]. Research showed that STAT3 deletion inhibits neutrophil chemotaxis and respiratory burst activity [50], while kaempferol can inhibit the expression of the JAK1/STAT3 signaling pathway in a variety of animal models [53,54]. Our results showed that the JAK1/STAT3 but not JAK2 signaling pathway was inhibited (western blot results shown in Appendix A), and neutrophil activation and infiltration decreased after 100 mg/kg kaempferol treatment. Therefore, it is reasonable to speculate that kaempferol inhibits activation of neutrophils in peripsheral blood and their infiltration into the ischemic brain by down-regulating the JAK1/STAT3 signaling pathway. However, the role of STAT3 in brain inflammation is still controversial, because STAT3 not only promotes cell death and leads to brain damage, but also is involved in neuronal survival [55]. Further experiments are needed to explore which aspect contributes more to the occurrence and development of cerebral ischemia-reperfusion injury.

In conclusion, the mechanism of kaempferol in the treatment of cerebral ischemic stroke by inhibiting apoptosis and regulating neuroinflammation involving neutrophils was investigated by network pharmacology analysis and experimental validation in vivo. The results demonstrated that kaempferol may reduce the number of apoptotic cells and regulate the expression of apoptosis-related proteins through the BDNF-TrkB-PI3K/AKT signaling pathway, and alleviated cerebral ischemia-reperfusion injury by inhibiting neutrophil activation, aggregation and infiltration and down-regulating JAK1/STAT3 signaling pathway. In conclusion, these results suggest that kaempferol is a promising drug for ischemic stroke treatment.

## 4. Materials and Methods

### 4.1. Network Pharmacology Analysis

#### 4.1.1. Prediction of Drug-Related Targets

The Traditional Chinese Medicine Systems Pharmacology Database and Analysis Platform (TCMSP, https://tcmsp-e.com/tcmsp.php (accessed on 4 September 2022), version 2.3), PubChem (https://pubchem.ncbi.nlm.nih.gov/ (accessed on 4 September 2022)) and Swiss Target Prediction databases (http://www.swisstargetprediction.ch/ (accessed on 4 September 2022)) were used to predict the targets of kaempferol. The parameter was set as “Homo sapiens” and probability attribute >0 was used as the screening standard. Then, all gene names were extracted from the uniport online protein database (UniProt, https://www.uniprot.org/ (accessed on 4 September 2022)) and named uniformly, so as to obtain the relevant targets of kaempferol.

#### 4.1.2. Collection of Targets of Ischemic Stroke

Then, we collected ischemic stroke-related targets via different databases to prepare for subsequent analysis. The keywords “ischemic stroke” and “cerebral infarction” were used in Online Mendelian Inheritance in Man (OMIM, https://omim.org/ (accessed on 4 September 2022)), DisGeNET (http://www.disgenet.org/ (accessed on 4 September 2022), version 7.0), GeneCards (https://www.genecards.org/ (accessed on 4 September 2022), version 5.12) and Therapeutic Target Database (TTD, http://db.idrblab.net/ttd/ (accessed on 4 September 2022)) to search for ischemic stroke-related targets. Then, the results of the 4 databases were summarized and the duplicates were deleted.

#### 4.1.3. PPI Network Construction

Common targets of kaempferol and ischemic stroke were screened and input into the STRING 11.5 database (https://cn.string-db.org/ (accessed on 4 September 2022)) to generate the PPI network. Cytoscape software was used for network visualization, and its NetworkAnalyzer plugin was used for parameter analysis.

#### 4.1.4. GO, KEGG Pathway and TRRUST Enrichment Analysis

To further determine possible functions and pathways of target genes, the common targets of kaempferol and ischemic stroke were uploaded to DAVID (https://david.ncifcrf.gov/ (accessed on 4 September 2022), version 2022q2). The select identifier was limited to “OFFICIAL GENE SYMBOL”, the list type was limited to “Gene List” and the species was limited to “Homo sapiens”. The obtained GO enrichment analysis result was presented by Excel software (version 2016), and the KEGG pathway enrichment analysis bubble chart was drawn using an online bioinformatics platform (https://www.omicshare.com/tools/Home/Soft/pathwaygsea (accessed on 4 September 2022)). Finally, eighteen core targets with the highest degree value of Cytoscape software were input into the Metascape database (https://metascape.org/ (accessed on 4 September 2022)) for enrichment analysis, which can obtain the transcriptional regulation information related to these targets.

### 4.2. Reagents and Antibody

Kaempferol was purchased from Chengdu Alfa biotechnology CO., LTD (Chengdu, China). Antibodies against caspase-3 (ab13847), BDNF (ab108319), TrkB (187041), COX-2 (ab15191), CitH3 (ab281584), Galectin-3 (ab76245) and ICAM-1 (ab171123) were purchased from Abcam (Cambridge, UK). Antibodies for β-actin (3700), p-PI3K (4288), PI3K (4257), Bcl-xl (2764T), p-AKT (9271S), AKT (9272S), MyD88 (4283), p-NF-κB (3033), NF-κB (4764), p-JAK1 (3331), JAK1 (88617), p-STAT3 (9145) and STAT3 (9139) were purchased from Cell Signaling Technology (Beverley, CA, USA). Anti-Bax antibody (50500-2-Ig), anti-Bcl-2 antibody (12789-1-AP), anti-ZO-1 antibody (1773-1-AP) and anti-occludin antibody (13409-1-AP) were the products of Proteintech (Wuhan, China). Antibodies for TLR4 (sc-293072) was provided by Santa Cruz Biotechnology Inc. (Santa Cruz, CA, USA). Antibodies against NeuN (GB13138-1), Iba-1 (GB13105-1) and GFAP (GB11096) were provided by Servicebio (Wuhan, China).

### 4.3. Animals and Experiment Design

Adult male Sprague–Dawley rats (8 weeks old, 280–300 g) were purchased from Laboratory Animal Centre of Beijing Hua-Fu-Kang Bioscience Co., Ltd. (Beijing, China; the animal certification number was SCXK (Jing) 2014-0004) and a total of 108 rats were used in this study (Appendix A). All animal care and experimental procedures were approved by the ethics committees of the Institute of Materia Medica, Chinese Academy of Medical Sciences & Peking Union Medical College. Animals were housed under standard temperature and humidity with a regular 12 h light-dark cycle and were allowed food and tap water ad libitum for three days prior to the start of the experiments. Based on previous research in our laboratory, cerebral ischemia for 90 min was selected to establish cerebral I/R rats model [56]. Rats were anesthetized by the intraperitoneal injection of sodium pentobarbital (30 mg/kg), and the rectal temperature was maintained at 37.0 ± 0.5 °C using heat pads during I/R period. Briefly, the right common carotid artery (CCA), external carotid artery (ECA), and internal carotid artery (ICA) were carefully separated. Then, the standardized nylon suture (0.38 ± 0.02 mm, 2838A4, Beijing Cinontech, Beijing, China) was slidden from ECA into ICA until the mark on the suture reached the intersection of ECA and ICA. After 90 min MCAO, the nylon suture was carefully removed for reperfusion. With the exception that the nylon suture was not inserted into ICA, the operation of sham group rats was the same as the above process.

After operation, rats were randomly divided into model group (I/R group) and kaempferol administration group (I/R + KAE group) according to random number table. In general, animal groups included in the study were (1) Sham group *n* = 28, (2) I/R group *n* = 28, (3) I/R + KAE 25 mg/kg group *n* = 12, (4) I/R + KAE 50 mg/kg group, *n* = 12 and (5) I/R + KAE 25 mg/kg group *n* = 28. Kaempferol was prepared in saline containing 0.5% sodium carboxymethylcellulose (CMC-Na) and delivered by intragastric at 2 h for the first time after reperfusion, followed by daily delivery until the endpoint. Rats were assessed for modified neurological severity score (mNSS) neurological function test 24 h after surgery, and the rats with mNSS score of 8–15 were included in our experiment. On day 3, rats were sacrificed for endpoint analyses except for peripheral blood neutrophils extracted on day 1 after I/R.

### 4.4. The mNSS Test

The mNSS test referred to previous studies [57], which reflected the motor, sensory, balance and reflex of rats, with a total score of 18 [58]. The higher the scores, the more serious was the injury.

### 4.5. Infarct Volume Assessments

Six rats in each group were sacrificed on day 3 to assess infract volume. After deep anesthesia, the brains were removed quickly and sliced into six 2 mm slices. Then, the sections were placed in 24-well plates and incubated at 37 °C with 2% 2, 3, 5-triphenyltetrazolium chloride (TTC, Sigma-Aldrich, St. Louis, MO, USA) for half an hour. They were then fixed with 4% PFA and photographed the next day. Infarct areas were measured using ImageJ software (National Institutes of Health (NIH), Bethesda, MD, USA, version 1.8.0.172). To rule out the possible interference of cerebral edema, infarct volume was calculated as follows: percentage of cerebral infarction volume (%) = (normal brain tissue area on the contralateral side of the infarction − (total area of the infarct side − infarct size))/(normal brain tissue area on the contralateral side of the infarction) × 100% [56].

### 4.6. Brain Edema Detection

Brain edema was assessed by calculating brain water content, which was defined as the weight difference between wet and dry samples [59]. After TTC staining, the infarct sides of brain slices were separated and weighed to obtain the wet weight, and then they were dried in an oven at 65 °C until the weight of the brain slices did not change within 1 h to obtain the dry weight. Brain water content of ischemic hemisphere was calculated as follow: wet to dry (W/D) ratio = ((wet weight − dry weight)/wet weight) × 100% [60].

### 4.7. TUNEL Staining

On the third day of reperfusion, four rats in each group were deeply anesthetized and cardially perfused with 0.9% saline and 4% PFA. Brains were removed, and coronal sections from −2.50 to −4.50 mm from bregma were fixed in 4% PFA for 24 h before being embedded in paraffin. Then, they were cut into 4 μm sections using a rotary microtome (RM2016, Leica, Wetzlar, Germany), dewaxed in xylene and dehydrated in alcohol. According to the manufacturer’s instructions, DNA damage was assessed by TUNEL Cell Apoptosis Detection Kit (Wuhan Servicebio technology, Wuhan, China) and 4,6-diamidino-2-phenylindole (DAPI) staining was used to count the total number of nuclei. The cortex and striatum of the ischemic cerebral hemisphere were divided into areas to be analyzed. The results of TUNEL staining were counted in three non-overlapping microscope fields and expressed as the number of positive cells (cells/mm^2^) using ImageJ software [31,61]. 

### 4.8. Immunofluorescence Detection of NeuN, Iba-1 and GFAP

The paraffin sections mentioned above were then incubated with anti-NeuN primary antibody (1:200), anti-Iba-1 primary antibody (1:200) and anti-GFAP primary antibody (1:800) overnight at 4 °C [61]. After washing with PBS, the brain sections were incubated with Cy3-conjugated secondary antibody (Wuhan Servicebio technology, Wuhan, China) for 2 h at room temperature and nuclei were stained with DAPI for 10 min at room temperature. The calculation method of immunofluorescence staining was similar to that of TUNEL-positive cells.

### 4.9. Evans Blue (EB) Leakage

On the third day of reperfusion, six rats in each group were injected with 4% EB dye dissolved in 0.9% saline into the tail vein (0.25 mL/100 g). After 2 h, the rats were anesthetized and perfused with 0.9% saline until the liquid flowing out of the right atrium was clear. Then, the brains were removed immediately and cut into 2.0 mm-thick sections. After taking photos, the brain slices were homogenized with 50% trichloroacetic acid (300 μL/100 mg) and centrifuged. The supernatant was collected and the absorbance value was measured (excitation wavelength 620 nm, emission wavelength 680 nm) [14,62].

### 4.10. Hematoxylin-Eosin (HE) Staining

The paraffin sections mentioned above were stained with hematoxylin for 3 min and eosin for 1 min. Finally, changes of brain tissues were observed under an optical microscope [63,64].

### 4.11. Determination of Myeloperoxidase (MPO) Activity

We took 4 mL blood from the abdominal aorta of rats mentioned in 4.5. and centrifuged at 4000 rpm at 4 °C for 15 min. The upper serum was carefully collected and stored at −80 °C until assay. MPO activity was measured using the MPO kit (Jian Cheng Bioengineering Institute, Nanjing, China) according to the recommended protocols [65]. An active unit of MPO is defined as degradation of 1 μM of hydrogen peroxide per minute at 37 °C, and the result of MPO activity was expressed as U/L [66].

### 4.12. Peripheral Blood Count

Six rats in each group were anesthetized at 24 h after reperfusion, and blood was collected from the abdominal aorta. Aliquots (20 μL) of blood were added to the EDTA-coated tubes containing isotonic diluents (2 mL, ISOTONAC-3), and then neutrophils were counted with an automatic blood analyzer (advia2120, Bayer, Tarrytown, NY, USA) [67].

### 4.13. Blood Neutrophil Isolation

We collected 10 mL blood from the abdominal aorta of rats mentioned in 4.12. and anticoagulated with heparin sodium. Neutrophils were isolated from peripheral blood using a Neutrophil Extraction Kit (Solarbio, Beijing, China) according to the manufacturer’s instructions.

### 4.14. Western Blotting Analysis

On the third day of reperfusion, cerebral ischemic cortices were removed from 6 rats in each group immediately after anesthesia and then stored at −80 °C for further detection. Peripheral blood neutrophils were extracted from peripheral blood mentioned in 4.13. using a neutrophil isolation kit (Solarbio, Wuhan, China). The total proteins were extracted with RIPA lysate containing cocktail protease inhibitor. Then, proteins were separated by sodium dodecyl sulfate-polyacrylamide gel electrophoresis (SDS-PAGE) and transferred to the PVDF membrane for Western blot analysis. The primary antibodies, which included anti-caspase-3, anti-Bax, anti-Bcl-2, anti-Bcl-xl, anti-BDNF-antibody, anti-TrkB-antibody, anti-p-PI3K antibody, anti-PI3K, anti-p-AKT antibody, anti-AKT, anti-COX-2, anti-TLR4, anti-MyD88, anti-NF-κB, anti-ZO-1 antibody, anti-occludin antibody, anti-phospho-JAK-1 antibody, anti-JAK-1 antibody, anti-phospho-STAT3, anti-STAT3 and anti-β-actin antibody, were incubated with membranes at 4 °C overnight. Then, the membranes were incubated with horseradish peroxidase-conjugated secondary antibody at room temperature for 2 h, and detected with an enhanced ECL system. The signal densities on the bolts were measured by Gel-Pro software (Molecular Imager ChemiDoc XRS + System, Bio-Rad, Irvine, CA, USA, version 4.0) and normalized by the value of anti-β-actin as an internal control (fold change relative to control) except for p-PI3K, p-AKT, p-NF-κB, p-JAK1 and p-STAT3 [68].

### 4.15. Real-Time PCR

The total RNA form ischemic cerebral cortex mentioned in 4.14 was extracted with Trizol reagent. Then, the total RNA (1 μg) was reverse-transcribed into cDNA using MonScriptTMR-TIII All-in-One Mix (Monad Biotech Co., Ltd., Wuhan, China). For real-time PCR, 2 μL cDNA templates were amplified with SYBR^®^ qPCR Master Mix (Vazyme Biotech Co., Ltd., Nanjing, China) using the following parameters: 5 min at 95 °C, followed by 40 cycles of 10 s at 95 °C, 30 s at 60 °C, and 15 s at 95 °C, 60 s for 60 °C and 15 s for 95 °C. According to the relative quantification of 2^−∆∆Ct^ method, the transcription levels of the target gene could be determined using β-actin as an internal reference [68]. The primers used in the experiment are listed in Table 1.

### 4.16. Statistical Analysis

All results are expressed as mean ± SEM. Data were assessed using one-way analysis of variance (ANOVA) followed by Dunnett’s multiple comparisons test. All statistical analyses were performed using GraphPad Prism software (GraphPad Software, San Diego, CA, USA, version 7.0).

## 5. Conclusions

In conclusion, our study identified the therapeutic mechanism of kaempferol on ischemic stroke through network pharmacology and verified it in in vivo experiments. We revealed that kaempferol regulates post-stroke apoptosis and neuroinflammation involved by neutrophils, and the underlying mechanism may be related to the regulation of BDNF-TrkB-PI3K/AKT signaling and JAK1/STAT3 signaling, respectively. With its protective potency in ischemic stroke, further clinical investigation into kaempferol as a promising therapeutic agent in patients with ischemic stroke is warranted.

## Figures and Tables

**Figure 1 ijms-23-12694-f001:**
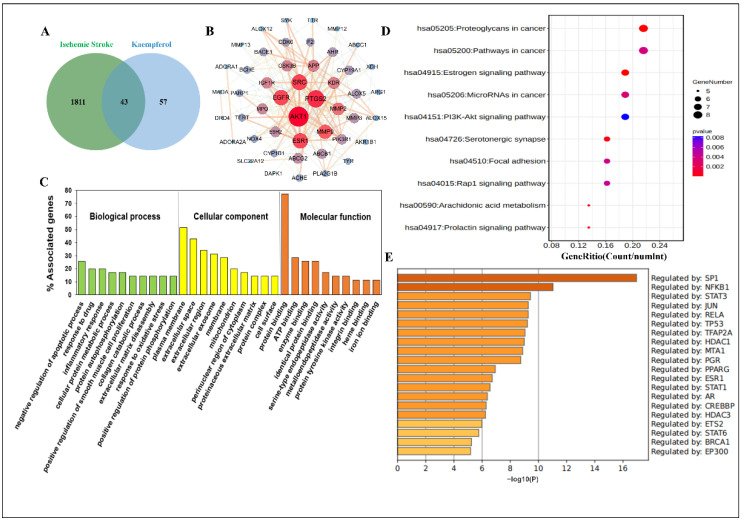
Protein–protein interaction (PPI) network and enrichment analysis of the anti-ischemic stroke mechanisms of kaempferol. (**A**) 43 common targets of kaempferol and ischemic stroke. (**B**) PPI network plotting of 36 main common targets by employing Cytoscape software. (**C**) The top 10 significantly enriched (*p* < 0.05) terms in BP, CC and MF of GO analysis. The *Y*-axis represents the enrichment count of the target, and the *X*-axis represents the GO category of the target gene. (**D**) The top 10 pathways with significantly enriched (*p* < 0.05) were selected. The *Y*-axis represents the main pathway, and the *X*-axis represents the gene ritio. (**E**) The transcription factor enrichment analysis of kaempferol against ischemic stroke in TRRUST using Metascape.

**Figure 2 ijms-23-12694-f002:**
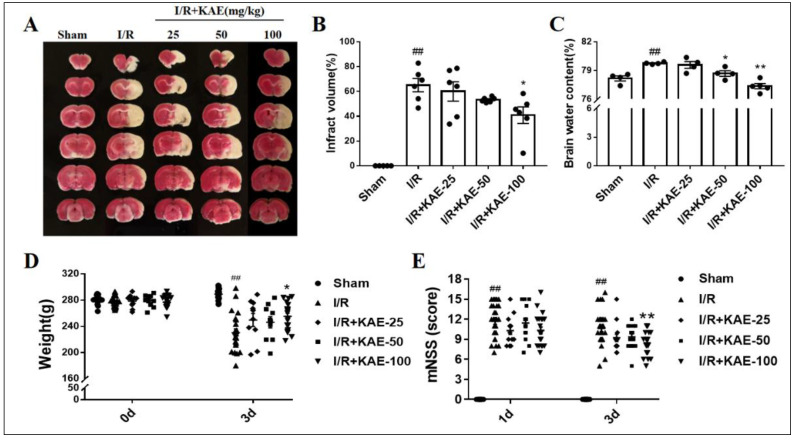
Effect of kaempferol on cerebral infarction volume, brain water content, weight and neurological function of rats on day 3 after cerebral ischemia-reperfusion. (**A**) Representative images of 2, 3, 5-triphenyltetrazolium chloride (TTC) staining. Statistical analysis of (**B**) infarct volume (*n* = 6), (**C**) brain water content (*n* = 6), (**D**) body weight (*n* = 10–12) and (**E**) modified neurological severity score (mNSS) score (*n* = 10–12). Values are expressed as mean ± SEM. ## *p* < 0.01 vs. Sham group; * *p* < 0.05, ** *p* < 0.01 vs. I/R group. KAE: kaempferol.

**Figure 3 ijms-23-12694-f003:**
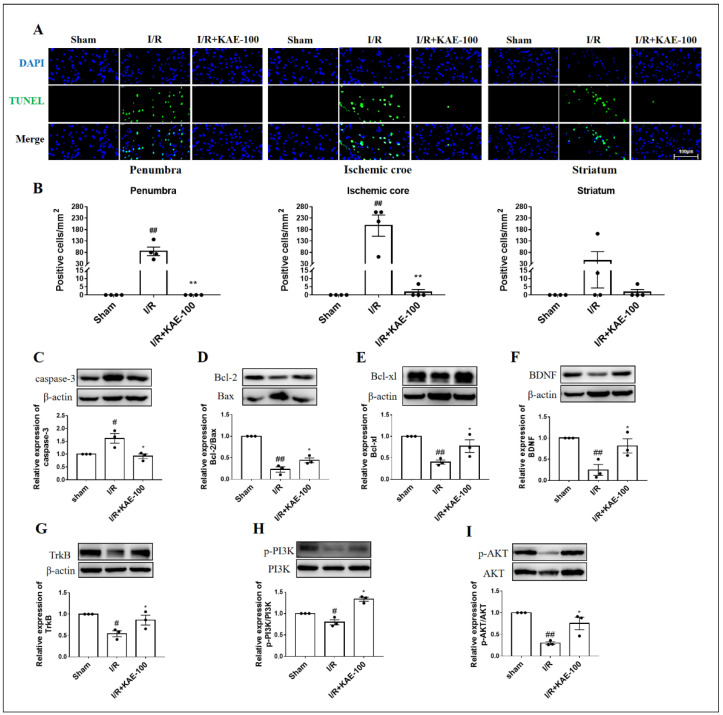
Effect of kaempferol on cell apoptosis of rats on day 3 after I/R. (**A**) Representative images of TUNEL staining (green) in penumbra. (**B**) Quantitative analysis of TUNEL-positive cells in penumbra, ischemic core and striatum (*n* = 4). The nuclei were stained with DAPI (blue), 630× magnification and scale bar = 100 μm. The expression levels of (**C**) caspase-3, (**D**) Bcl-2/Bax, (**E**) Bcl-xl, (**F**) BDNF, (**G**) TrkB, (**H**) p-PI3K and (**I**) p-AKT were assessed by Western blot analysis. Values are expressed as mean ± SEM (*n* = 3) # *p* < 0.05, ## *p* < 0.01 vs. Sham group; * *p* < 0.05, ** *p* < 0.01 vs. I/R group. KAE: kaempferol.

**Figure 4 ijms-23-12694-f004:**
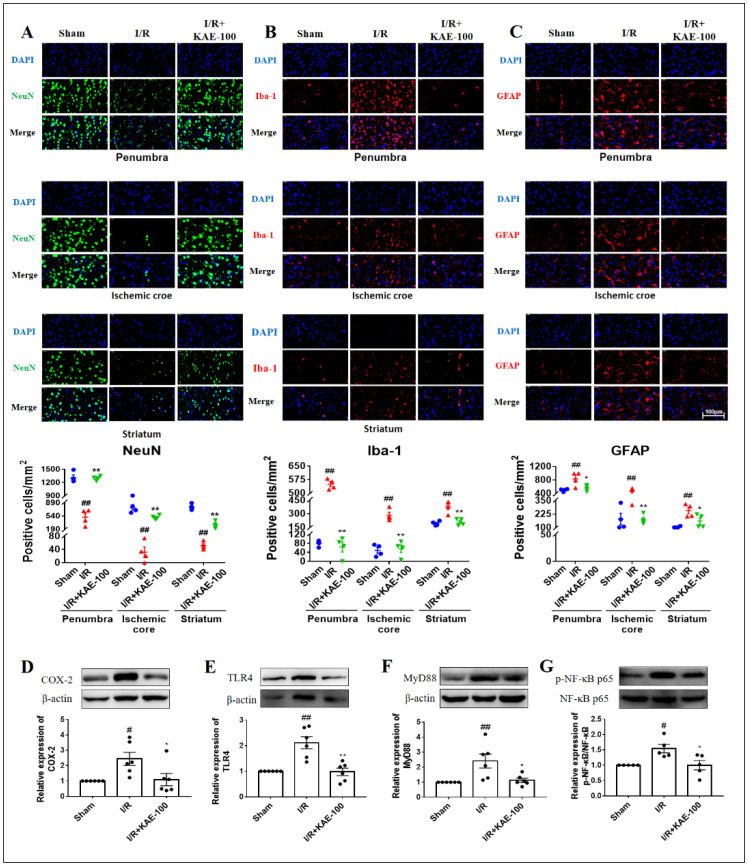
Effect of kaempferol on neuron loss and glial activation on day 3 after cerebral I/R. Representative fluorescent micrographs of (**A**) NeuN, (**B**) Iba-1 and (**C**) GFAP in penumbra, ischemic core and striatum and quantitative analysis of immunofluorescent staining (*n* = 4). 630× magnification, scale bar = 100 μm. The expression levels of (**D**) COX-2, (**E**) TLR4, (**F**) MyD88 and (**G**) p-NF-κB p65 were assessed by Western blot. Values are expressed as mean ± SEM (*n* = 5–6) # *p* < 0.05, ## *p* < 0.01 vs. Sham group; * *p* < 0.05, ** *p* < 0.01 vs. I/R group. KAE: kaempferol.

**Figure 5 ijms-23-12694-f005:**
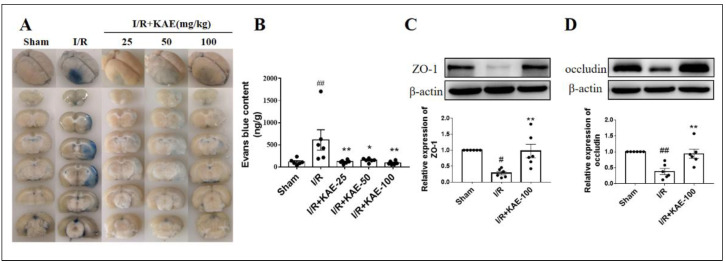
Effect of kaempferol on BBB disruption of rats on day 3 after cerebral I/R. (**A**) Representative images of Evans blue staining. (**B**) Quantitative analysis of Evans blue content in ischemic hemisphere. The expression levels of (**C**) ZO-1 and (**D**) occludin were assessed by Western blot analysis. Values are expressed as mean ± SEM (*n* = 6). # *p* < 0.05, ## *p* < 0.01 vs. Sham group; * *p* < 0.05, ** *p* < 0.01 vs. I/R group. KAE: kaempferol.

**Figure 6 ijms-23-12694-f006:**
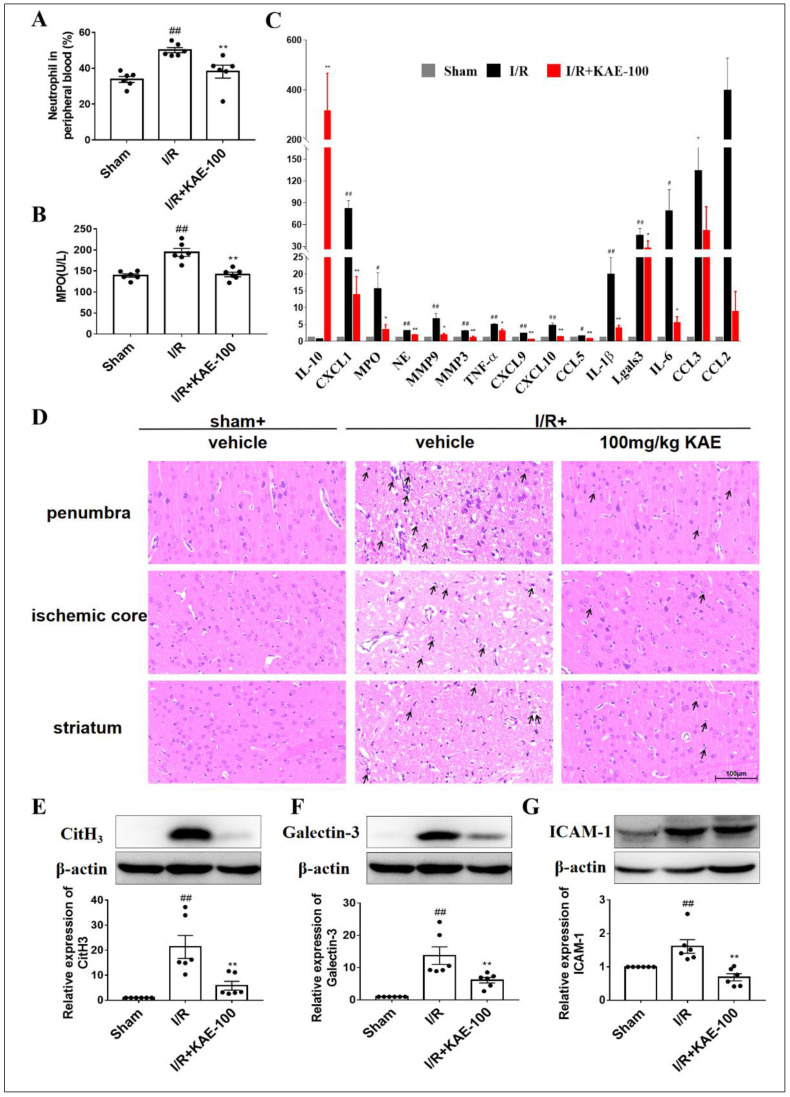
Effect of kaempferol on activation, aggregation and infiltration of neutrophils in rats on day 3 after cerebral I/R. (**A**) Neutrophil count in the peripheral blood was detected by Blood Analyzer. (**B**) MPO activity in serum was detected by ELISA kit. (**C**) Transcription levels of protease, chemokines and inflammatory factors released by neutrophils were detected by RT-PCR (*n* = 6). (**D**) Representative images of HE staining in penumbra, ischemic core and striatum. 400× magnification and scale bar = 100 μm. Black arrows indicate neutrophils (*n* = 4). Expression levels of neutrophil related proteins (**E**) CitH3, (**F**) Galectin-3 and (**G**) ICAM-1 in ischemic cortex were detected by Western blotting. Values are expressed as mean ± SEM (*n* = 6). # *p* < 0.05, ## *p* < 0.01 vs. Sham group; * *p* < 0.05, ** *p* < 0.01 vs. I/R group. KAE: kaempferol.

**Figure 7 ijms-23-12694-f007:**
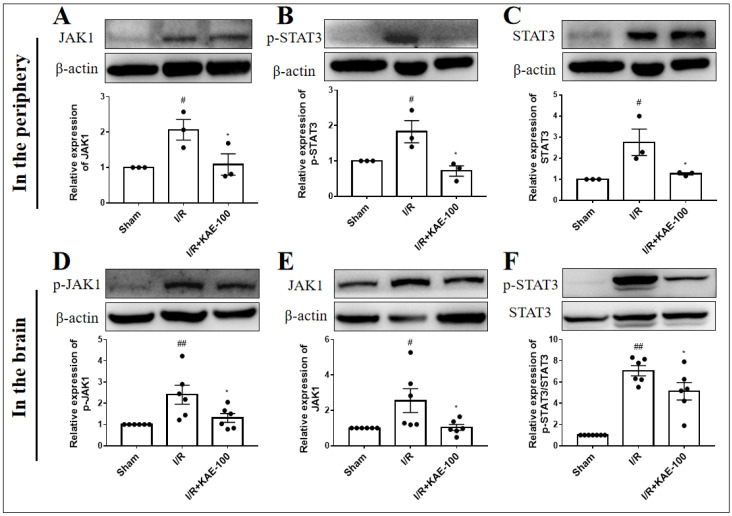
Effect of 100 mg/kg kaempferol on the expression of JAK1/STAT3 signaling pathway in peripheral blood neutrophils of rats on day 1 and in ischemic cortex of rats on day 3 after cerebal I/R. The expression level of (**A**) JAK1, (**B**) p-STAT3, (**C**) STAT3 in peripheral blood neutrophils (*n* = 3) and (**D**) p-JAK1, (**E**) JAK1, (**F**) p-STAT3 in ischemic cortex was assessed by Western blotting analysis. Values are expressed as mean ± SEM. # *p* < 0.05, ## *p* < 0.01 vs. Sham group; * *p* < 0.05 vs. I/R group. KAE: kaempferol.

**Table 1 ijms-23-12694-t001:** Primers used in this study.

Gene	Forward Primer (5′-3′)	Reverse Primer (5′-3′)	Reference
IL-10	CCTCTGGATACAGCTGCGAC	ATGGCCTTGTAGACACCTTTGT	[69]
CXCL1	AGACAGTGGCAGGGATTCAC	AAGCCTCGCGACCATTCTTG	[70]
MPO	ATGTCACAGGGGACATGCG	TCTGTGGCCAGACGGTTATG	[71]
NE	CTCCGTGGCCAACGATAAGA	AGGTCACTCGTCCTACAGGT	[72]
MMP9	GCAAACCCTGCGTATTTCCATT	GCGATAACCATCCGAGCGAC	[73]
MMP3	CATGAACTTGGCCACTCCCT	TGGGTACCACGAGGACATCA	[73]
Arg-1	ACAAGACAGGGCTACTTTCAGG	ACAAGACAAGGTCAACGCCA	[74]
TNF-α	CTGTGCCTCAGCCTCTTCTC	ACTGATGAGAGGGAGCCCAT	[75]
CXCL9	TGCCTAGACCCAGATTCAGC	AGATGCAGAGCGCTTGTTGG	[70]
CXCL10	TTATTGAAAGCGGTGAGCCAAAG	GCTGTCCATCGGTCTCAGCA	[76]
CCL5	CGTGAAGGAGTATTTTTACACCAGC	CTTGAACCCACTTCTTCTCTGGG	[77]
IL-1β	CCCTGAACTCAACTGTGAAATAGCA	CCCAAGTCAAGGGCTTGGAA	[76]
Lgals3	CAACTGGCCCTAGTGCTTATC	CAGAGTGATACTGTTTGCGTTG	[78]
IL-6	GCCCACCAGGAACGAAAGTC	GGCTGGAAGTCTCTTGCGGA	[79]
CCL3	TGCCCTTGCTGTTCTTCTCT	AAAGGCTGCTGGTCTCAAAA	[80]
CCL2	ATGCAGTTAATGCCCCACTC	TTCCTTATTGGGGTCAGCAC	[80]
GAPDH	AAGTTCAACGGCACAGTCAAG	ACATACTCAGCACCAGCATCA	[81]

## Data Availability

Not applicable.

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
