# Peer review of "Network Pharmacology Analysis and Experimental Validation of Kaempferol in the Treatment of Ischemic Stroke by Inhibiting Apoptosis and Regulating Neuroinflammation Involving Neutrophils"

_ijms, 2022, doi:10.3390/ijms232012694_

Round 1
Reviewer 1 Report
This is an interesting and well done study examining the effect of kaempferol in the treatment of ischemic stroke.
Comments:
Major comments: more details are needed in the animal methods. Age? # per experiment?
Data with bar graphs should show actual data points not a solid histogram.
The supplementary westerns only show a sliver of the blot so I am not sure why they are included.
Everything changed/improved. They should show at least one protein that did not change with the intervention.
Author Response
Point 1: more details are needed in the animal methods. Age? # per experiment?
Response 1: Thanks for your question. The age and the number of animals in each experiment were added in revised manuscript and Supplementary information (Figure S1).
After revise:
Page13 line 463-466: Adult male Sprague-Dawley rats (8 weeks old, 280-300g) were purchased from Laboratory Animal Centre of Beijing Hua-Fu-Kang Bioscience Co., Ltd. (Beijing, China; the animal certification number was SCXK (Jing) 2014-0004) and a total of 108 rats were used in this study.
Page13 line 480-484: After operation, rats were randomly divided into model group (I/R group) and kaempferol administration group (I/R + KAE group) according to random number table. In general, animal groups included in the study were (1) Sham group n=28, (2) I/R group n = 28, (3) I/R + KAE 25 mg/kg group n = 12, (4) I/R + KAE 50 mg/kg group, n = 12 and (5) I/R + KAE 25 mg/kg group n = 28.
Page14 line 495-497:
4.5. Infarct volume assessments
Six rats in each group were sacrificed on day 3 to assess infract volume. After deep anesthesia, the brains were removed quickly and sliced into six 2 mm slices.
Page14 line 506-511:
4.6. Brain edema detection
Brain edema was assessed by calculating brain water content, which was defined as the weight difference between wet and dry samples [59]. After TTC staning, the infarct side of brain slices were separated and weighed to obtain the wet weight, and then they were dried in an oven at 65 ℃ until the weight of the brain slices did not change within 1 h to obtain the dry weight.
Page14 line 513-517:
4.7. TUNEL staining
On the third day of reperfusion, four rats in each group were deeply anesthetized and cardially perfused with 0.9% saline and 4% PFA. Brains were removed, and coronal sections from -2.50 to -4.50 mm from bregma were fixed in 4% PFA for 24 h before embedded in paraffin.
Page14 line 526-529:
4.8. Immunofluorescence detection of NeuN, Iba-1 and GFAP
The paraffin sections mentioned above were then incubated with anti-NeuN primary antibody (1:200), anti-Iba-1 primary antibody (1:200) and anti-GFAP primary antibody (1:800) overnight at 4 °C [61].
Page14-15 line 534-536:
4.9. Evans blue (EB) leakage
On the third day of reperfusion, six rats in each group were injected with 4% EB dye dissolved in 0.9% saline into the tail vein (0.25mL/100g).
Page15 line 542-545:
4.10. Hematoxylin-Eosin (HE) staining
The paraffin sections mentioned above were stained with hematoxylin for 3min and eosin for 1 min. Finally, changes of brain tissues were observed under an optical microscope [63,64].
Page15 line 546-548:
4.11. Determination of myeloperoxidase (MPO) activity
4 mL blood was taken from the abdominal aorta of rats mentioned in 4.5. and centrifuged at 4000 rpm at 4 ℃ for 15 min.
Page15 line 553-555:
4.12. Peripheral blood count
Six rats in each group were anesthetized at 24 h after reperfusion, and blood was collected from the abdominal aorta.
Page15 line 559-561:
4.13. Blood neutrophil isolation
10 mL blood was collected from abdominal aorta of rats mentioned in 4.12. and anticoagulated with heparin sodium.
Page15 line 564-569:
4.14. Western blotting analysis
On the third day of reperfusion, cerebral ischemic cortices were removed from 6 rats in each group immediately after anesthesia and then stored at −80 ℃ for further detection. Peripheral blood neutrophils were extracted from peripheral blood mentioned in 4.13. using a neutrophil isolation kit (Solarbio, Wuhan, China). The total pro-teins were extracted with RIPA lysate containing cocktail protease inhibitor.
Page16 line 582-584:
4.15. Real-time PCR
Total RNA form ischemic cerebral cortex mentioned in 4.14 was extracted with Trizol reagent.
Point 2: Data with bar graphs should show actual data points not a solid histogram.
Response 2: Thanks for your suggestion. We have modified the Figures with actual data points according your suggestion.
After revise:
Page 4: Figure 2
Page 5: Figure 3
Page 6: Figure 4
Page 7: Figure 5
Page 8: Figure 6
Page 9: Figure 7
Point 3: The supplementary westerns only show a sliver of the blot so I am not sure why they are included.
Response 3: Thanks for your question. Although on the same membrane, only the framed band is our target band, the other bands do not belong to kaempferol. In further experiments, we will distinguish the samples of these different drugs to avoid misunderstanding.
Point 4: Everything changed/improved. They should show at least one protein that did not change with the intervention.
Response 4: Thanks for your question. We performed WB analysis on many proteins of interest, and the article showed the proteins with statistical differences. During the experiment, some proteins with little change were also found, such as p-JAK-2, PAD4, iNOS, integrin β2, MMP-2 and MMP-3. The statistical results are shown in supplementary information (Figure S2).
Reviewer 2 Report
The study by Shan-Shan Zhang et al. investigates the potential mechanism of the neuroprotective effect of kaempferol, a bioactive substance, on the degree of ischemic injury in a rat MCAO model . The study uses a variety of techniques including neurological scores assessment, TTC, H&E, and IHC, RT-PCR, and WB. In general, the work makes a good impression, but the manuscript contains a lot of omissions that should be corrected by the authors.
Introduction section
1. The authors write: “Kaempferol has been found to have neuroprotective effects by reducing oxidative stress, neuroinflammation and mitochondrial dysfunction after brain IRI”. This key statement, important for the study, does not support by the reference on a previously published paper.
Method section
2. Experimental design in Method section does not specify the time point, or points, after MCAO when rats were sacrificed. Only from the results can one guess that the time points were 1 and 3 days after
MCAO. You should specify this both in the 'Experimental design' section and for each type of analysis.
3. It is not specified kaempferol doses and the route of administration. The authors only write "kaempferol of different concentrations".
4. The Method section does not specify a sample size for each type of analysis (evaluation of infarct volume size, IHC, RT-PCR, brain edema assessment etc). The total sample size, as well as the
number of animals in each group, is also not specified.
5. Section “Reagents and Antibody”. Please, check carefully that you described all antibodies. At least, I do not see antibodies for immunochemistry (Iba1, GFAP, NeuN).
6. The details of cell counting in image analysis are very unclear. The authors wrote: “The cortex and striatum of ischemic cerebral hemisphere were divided into areas to be analyzed.” How were the zones of the ischemic core and penumbra divided? In what part of the striatum, with the exception of the core and penumbra, were the calculations performed? It would be good to provide an illustration with a calculation scheme.
7. How many brain slices were analyzed?
8. Real-time PCR. It is not specified how the animals were killed to collect the brains.
Results
9. The results give the impression that the animals of the kaempferol+ischemia group do not differ at all from the sham-operated animals in terms of the number of neurons, astrocytes, and microglia. However, Figure 2a clearly shows that there is still an ischemic core, which means that there are changes in the nervous tissue. Perhaps the problem is that the cell counts were carried out outside the ischemic core, on the slices without lesion?
10. Figure 4c. Does the number of GFAP+ cells in the ischemic core in the I/R group differ from the sham animals?
Author Response
Point 1: Introduction section: The authors write: “Kaempferol has been found to have neuroprotective effects by reducing oxidative stress, neuroinflammation and mitochondrial dysfunction after brain IRI”. This key statement, important for the study, does not support by the reference on a previously published paper.
Response 1: Thanks for your advice. The supporting reference have added at the appropriate position of the manuscript.
After revise:
Page2 line 69-70: Kaempferol has been found to have neuroprotective effects by reducing oxidative stress [13], neuroinflammation [14] and mitochondrial dysfunction [15] after brain IRI.
Point 2: Experimental design in Method section does not specify the time point, or points, after MCAO when rats were sacrificed. Only from the results can one guess that the time points were 1 and 3 days after MCAO. You should specify this both in the 'Experimental design' section and for each type of analysis.
Response 2: Thanks for your advice. We added more description of the time point.
After revise:
Page13-14 line 488-490: On day 3 rats were sacrificed for endpoint analyses except for peripheral blood neutrophils extracted on day 1 after I/R.
Point 3: It is not specified kaempferol doses and the route of administration. The authors only write "kaempferol of different concentrations".
Response 3: Thanks for your advice. We added more description of kaempferol doses and the route of administration.
After revise:
Page13 line 480-486: After operation, rats were randomly divided into model group (I/R group) and kaempferol administration group (I/R + KAE group) according to random number table. In general, animal groups included in the study were (1) Sham group n=28, (2) I/R group n = 28, (3) I/R + KAE 25 mg/kg group n = 12, (4) I/R + KAE 50 mg/kg group, n = 12 and (5) I/R + KAE 25 mg/kg group n = 28. Kaempferol was prepared in saline containing 0.5% sodium carboxymethylcellulose (CMC-Na) and delivered by intragastric at 2 h for the first time after reperfusion, followed by daily delivery until the endpoint.
Point 4: The Method section does not specify a sample size for each type of analysis (evaluation of infarct volume size, IHC, RT-PCR, brain edema assessment etc). The total sample size, as well as the number of animals in each group, is also not specified.
Response 4: Thanks for your advice. The exact number of rats used in each experiment is summarized in the Supplementary information (Figure S1) and we also added more description of sample size for each type of analysis, total sample size and the number of animals in each group.
After revise:
Page14 line 495-497:
4.5. Infarct volume assessments
Six rats in each group were sacrificed on day 3 to assess infract volume. After deep anesthesia, the brains were removed quickly and sliced into six 2 mm slices.
Page14 line 506-511:
4.6. Brain edema detection
Brain edema was assessed by calculating brain water content, which was defined as the weight difference between wet and dry samples [59]. After TTC staning, the infarct side of brain slices were separated and weighed to obtain the wet weight, and then they were dried in an oven at 65 ℃ until the weight of the brain slices did not change within 1 h to obtain the dry weight.
Page14 line 513-517:
4.7. TUNEL staining
On the third day of reperfusion, four rats in each group were deeply anesthetized and cardially perfused with 0.9% saline and 4% PFA. Brains were removed, and coronal sections from -2.50 to -4.50 mm from bregma were fixed in 4% PFA for 24 h before embedded in paraffin.
Page14 line 526-529:
4.8. Immunofluorescence detection of NeuN, Iba-1 and GFAP
The paraffin sections mentioned above were then incubated with anti-NeuN primary antibody (1:200), anti-Iba-1 primary antibody (1:200) and anti-GFAP primary antibody (1:800) overnight at 4 °C [61].
Page14-15 line 534-536:
4.9. Evans blue (EB) leakage
On the third day of reperfusion, six rats in each group were injected with 4% EB dye dissolved in 0.9% saline into the tail vein (0.25mL/100g).
Page15 line 542-545:
4.10. Hematoxylin-Eosin (HE) staining
The paraffin sections mentioned above were stained with hematoxylin for 3min and eosin for 1 min. Finally, changes of brain tissues were observed under an optical microscope [63,64].
Page15 line 546-548:
4.11. Determination of myeloperoxidase (MPO) activity
4 mL blood was taken from the abdominal aorta of rats mentioned in 4.5. and centrifuged at 4000 rpm at 4 ℃ for 15 min.
Page15 line 553-555:
4.12. Peripheral blood count
Six rats in each group were anesthetized at 24 h after reperfusion, and blood was collected from the abdominal aorta.
Page15 line 559-561:
4.13. Blood neutrophil isolation
10 mL blood was collected from abdominal aorta of rats mentioned in 4.12. and anticoagulated with heparin sodium.
Page15 line 564-569:
4.14. Western blotting analysis
On the third day of reperfusion, cerebral ischemic cortices were removed from 6 rats in each group immediately after anesthesia and then stored at −80 ℃ for further detection. Peripheral blood neutrophils were extracted from peripheral blood mentioned in 4.13. using a neutrophil isolation kit (Solarbio, Wuhan, China). The total pro-teins were extracted with RIPA lysate containing cocktail protease inhibitor.
Page16 line 582-584:
4.15. Real-time PCR
Total RNA form ischemic cerebral cortex mentioned in 4.14 was extracted with Trizol reagent.
Page13 line 463-466: Adult male Sprague-Dawley rats (8 weeks old, 280-300g) were purchased from Laboratory Animal Centre of Beijing Hua-Fu-Kang Bioscience Co., Ltd. (Beijing, China; the animal certification number was SCXK (Jing) 2014-0004) and a total of 108 rats were used in this study.
Page13 line 482-484: In general, animal groups included in the study were (1) Sham group n=28, (2) I/R group n = 28, (3) I/R + KAE 25 mg/kg group n = 12, (4) I/R + KAE 50 mg/kg group, n = 12 and (5) I/R + KAE 25 mg/kg group n = 28.
Point 5: Section “Reagents and Antibody”. Please, check carefully that you described all antibodies. At least, I do not see antibodies for immunochemistry (Iba1, GFAP, NeuN).
Response 5: Thanks for your advice. We carefully checked all antibody information and supplemented the missing antibody information in the corresponding position of the manuscript.
After revise:
Page13 line 459-461: Antibodies against NeuN (GB13138-1), Iba-1 (GB13105-1) and GFAP (GB11096) were provided by Servicebio (Wuhan, China).
Point 6: The details of cell counting in image analysis are very unclear. The authors wrote: “The cortex and striatum of ischemic cerebral hemisphere were divided into areas to be analyzed.” How were the zones of the ischemic core and penumbra divided? In what part of the striatum, with the exception of the core and penumbra, were the calculations performed? It would be good to provide an illustration with a calculation scheme.
Response 6: Thanks for your suggestion. After TTC staining or immunofluorescence, the respective positions of ischemic core and penumbra can be clearly seen on brain slices. According to the localization map of rat brain, we provided the schematic diagram of ischemic core area, peripheral area and striatum on the ischemic side seen in Figure 3A(a).
After revise:
Page 5: Figure 3. Effect of kaempferol on cell apoptosis of rats on day 3 after I/R. (A) The results of TUNEL staining. (a)Representative images of TUNEL staining (green) in penumbra. (b) Quantitative analysis of TUNEL-positive cells in penumbra, ischemic core and striatum (n = 4). The nuclei were stained with DAPI (blue), 630 × magnification and scale bar=100μm. (c) The location of markers used for statistics in rat brain was displayed. (â… ) ischemic penumbra, (â…¡) ischemic core and (â…¢) Striatum. The expression levels of (B) caspase-3, (C) Bcl-2/Bax, (D) Bcl-xl, (E) BDNF, (F) TrkB, (G) p-PI3K and (H) p-AKT were assessed by Western blot analysis. Values are expressed as mean ± SEM (n = 3) # p <0.05, ## p <0.01 vs. Sham group; * p <0.05 vs. I/R group. KAE: kaempferol.
Point 7: How many brain slices were analyzed?
Response 7: Thanks for your question. For immunofluorescence detection, a total of 48 brain slices were analyzed. In addition, the number of TUNEL/NeuN/Iba-1/GFAP positive cells in ischemic core, penumbra and striatum was counted in three different fields of view in each brain slice. Finally, the average value of three counts was used for statistical analysis.
Point 8: Real-time PCR. It is not specified how the animals were killed to collect the brains.
Response 8: Thanks for your advice. For a clearer description, we changed the order of Real-time PCR and Western blotting analysis in the method section, and we added more description about how the rats were killed to collect the brains in 4.14. Western blotting analysis.
After revise:
Page15 line 564-569:
4.14. Western blotting analysis
On the third day of reperfusion, cerebral ischemic cortices were removed from 6 rats in each group immediately after anesthesia and then stored at −80 ℃ for further detection. Peripheral blood neutrophils were extracted from peripheral blood mentioned in 4.13. using a neutrophil isolation kit (Solarbio, Wuhan, China). The total pro-teins were extracted with RIPA lysate containing cocktail protease inhibitor.
Page16 line 582-584:
4.15. Real-time PCR
Total RNA form ischemic cerebral cortex mentioned in 4.14 was extracted with Trizol reagent.
Point 9: The results give the impression that the animals of the kaempferol+ischemia group do not differ at all from the sham-operated animals in terms of the number of neurons, astrocytes, and microglia. However, Figure 2a clearly shows that there is still an ischemic core, which means that there are changes in the nervous tissue. Perhaps the problem is that the cell counts were carried out outside the ischemic core, on the slices without lesion?
Response 9: Thanks for your question. The coronal section with a distance of -2.50 to -4.50 mm from bregma is the section where we do immunofluorescence, which indeed corresponds to the fourth to fifth brain slices in the TTC section. However, TTC staining showed that in the I/R group, there was obvious ischemic injury at this location, which was alleviated by kaempferol administration, so that the ischemic injury might not be observed. Meanwhile, the penumbra is the most valuable region for rescue after ischemia. The results of immunofluorescence staining showed that kaempferol seemed to alleviate ischemic injury in the penumbra more obviously, which made it seem that there was no difference between the I/R + kaempferol group and the sham group. In fact, based on the immunofluorescence results of penumbra, ischemic core and striatum, kaempferol does have a very significant therapeutic effect, but more indicators need to be detected in further experiments.
Point 10: Figure 4c. Does the number of GFAP+ cells in the ischemic core in the I/R group differ from the sham animals?
Response 10: Thanks for your question. The representative immunofluorescence images of GFAP+ staining was added in Figure 4.
Round 2
Reviewer 1 Report
Improved manuscript.
Still needs editing for English and typos.
Author Response
We corrected the English and typing errors in the manuscript again.
Reviewer 2 Report
The authors worked hard to correct the article and removed a number of questions that arose. However, one significant question regarding the effect of kaempferol on neuronal loss, inflammation, and astrogliosis (point 9) remained. In our experience, the ischemic lesion in the MCAO model can differ significantly in size and location for different animals even without therapy. The correspondence of coordinates of TTC-stained sections and sections for immunohistochemistry does not yet mean that the core was located exactly there and that it was present at all. Sometimes operations are less successful and, for example, there is no complete interruption of blood flow in the circle of Willis. The criterion for the success of blocking the blood flow can be MRI or angiography, measuring local blood flow by laser-Doppler flowmetry in vivo, or the presence of a lesion on postmortem sections. How can the authors confirm the presence of ischemia in animals whose sections were examined immunohistochemically? Indeed, in these animals, the lesion was not actually identified (no neuronal loss, inflammation, and astrogliosis). Is this the result of a failed operation? The presence of a lesion according to TTC staining and the absence of neuronal loss, inflammation, and astrogliosis (i.e., the absence of ischemic core) directly contradict each other. This contradiction needs to be resolved.
Author Response
Thanks for your valuable comments and questions. "Even without treatment, there may be significant differences in the size and location of ischemic lesions in MCAO models of different animals", which is a common problem in the application of animal models to drug efficacy evaluation, that is, the model group and the drug administration group may have the possibility of self-recovery. Therefore,statistical analysis between the drug administration group and the model group is required in the efficacy evaluation.
For the application of MCAO model in drug evaluation, the technical proficiency of operators really has a great impact, and it is very important to ensure the success of the MCAO model. Our research team has matured technical methods and skilled operators to prepare MCAO rats’ model. In order to establish a stable MCAO rat model, our previous research have applied the detection laser Doppler flowmetry, MRI, HE staining, Nissl’s staining, immunohistochemistry and behavioral tests to ensure the success of the model. In the present study, HE staining, immunohistochemistry and behavioral tests were applied to evaluate kaempferol treatment on cerebral ischemia. From the HE staining, the location of (a) penumbra, (b) ischemic core and (c) striatum in ischemic brain was observed (Fig. S1).